# Molecular Dynamics Study on Mechanical Properties of Cellulose with Water Molecules Diffusion Behavior at Different Oxygen Concentrations

**Yuanyuan Guo, Wei Wang * and Xuewei Jiang**

College of Engineering and Technology, Northeast Forestry University, Harbin 150040, China
* Correspondence: vickywong@nefu.edu.cn; Tel.: +86-133-1361-3588

**Abstract:** Six groups of cellulose-water-oxygen simulation models with oxygen concentrations of 0%, 2%, 4%, 6%, 8%, and 10% were established by molecular dynamics software to analyze the effect of oxygen concentration on the mechanical properties of wood cellulose during water vapor heat treatment in terms of the number of hydrogen bonds, the diffusion coefficient of water molecules, the mean square displacement of cellulose chains, and mechanical parameters. The results showed that the diffusion coefficient of water molecules increased steadily as oxygen concentration increased, which affected cell size and density to some extent. The mean square displacement of the cellulose chain at a higher oxygen concentration was larger than at a lower oxygen concentration, indicating that the cellulose chain became more unstable at high oxygen concentration. This trend was consistent with the amount of hydrogen bonds inside the cellulose chains. The analysis of mechanical parameters showed that Young's modulus and shear modulus showed a trend of increasing and then decreasing with increasing oxygen concentration, and wood cellulose had good resistance to deformation and rigidity at 2% oxygen concentration. Therefore, during the heat treatment of wood, appropriately increasing the oxygen concentration will potentially improve the rigidity and distortion resistance of wood.

**Keywords:** water vapor heat treatment; molecular dynamics; oxygen concentration; wood cellulose; mechanical properties





## 1. Introduction

As a renewable resource, wood provides great convenience for human use because of its light texture and beautiful appearance. Wood can be used as a material for interior decoration and furniture manufacturing, and because of its diversity of design styles, it greatly enriches the choice of wood products. Heat treatment is the most industrially successful and economically efficient method of wood modification [1]. However, the requirements for heat treatment of wood are very high. If care is not taken, deformation or cracking will occur, which affects the use of wood [2]. Therefore, reasonable process optimization of wood has great significance and value.

The hygroscopicity of wood can be effectively reduced by heat treatment technology. Zhou et al. investigated the hygroscopicity of wood by measuring the wood-water related parameters of heat-treated lumber thermally and showed that heat treatment lowered the fiber saturation point of wood, resulting in a decrease in the hygroscopicity of heat-treated wood [3]. Fu et al. investigated the effects of heat treatment on wood moisture absorption and other water-related properties and found that surface wettability was significantly weakened after heat treatment, which was manifested as an increased contact angle and decreased surface free energy [4]. Heat treatment technology also makes the wood more dimensionally stable, more uniform in color, and longer-lasting, but it can also lead to the destruction of the mechanical properties of the wood [5–8]. Bruno et al. found that the dimensional stability of *Paulownia tomentosa (Thunb.) Steud.* wood improved after heat

treatment, but its mechanical qualities deteriorated [9]. Lee et al. examined the physical and mechanical properties of particleboards made from heat-treated rubberwood particles and found that heat treatment increased the dimensional stability of particleboards but had a negative effect on their mechanical quality [10]. Lu et al. studied the chemical changes of the heat treatment parameters (temperature and duration) on the surface color and gloss of young eucalyptus lumber. The outcomes showed that the temperature of the heat treatment had a substantial impact on the coloring properties of young eucalyptus lumber [11].

As the most common heat treatment medium in China, superheated steam is more practical, economical, and convenient than other media (inert gas, air, and hot oil). Many water vapor heat treatment processes are carried out at a certain oxygen concentration, which increases the processing efficiency of wood heat treatment to some extent. However, fire incidents occur in the heat treatment kiln. This is likely because the oxygen concentration in the heat treatment kiln is too high. Therefore, the critical value of oxygen is set at 10% in this paper. During superheated steam heat treatment, the oxygen concentration in the kiln has an impact on the mechanical properties of the heat-treated material [12]. In this paper, when the simulation results of each model are compared with the non-oxygen model, it is found that wood cellulose has good resistance to deformation and rigidity at oxygen concentrations below 6%. However, these properties are significantly diminished at oxygen concentrations higher than 6%.

Cellulose makes up a large portion of wood. Natural cellulose is a macromolecular polysaccharide that is composed of interwoven, overlapping crystalline and amorphous regions [13]. The arrangement between cellulose molecules in the crystalline region is very compact and orderly. During water vapor heat treatment, water molecules are limited to the surface of the crystalline area, so they do not interact well with the crystalline area [14]. In the amorphous region, the molecular structure between celluloses is disordered [15], which has a strong adsorption effect on water molecules. This results in the structure between celluloses to become easily destroyed by water molecules. Therefore, this study carefully investigates the effect of the interaction between the amorphous region and water vapor on the heat treatment of wood at different oxygen concentrations.

Nowadays, analog computing techniques have gained prominence. The use of molecular simulations to derive the microscopic properties of molecules and predict the microscopic, mesoscopic, and macroscopic properties of products and materials has become an emerging theoretical development trend. In this paper, simulations are performed using Materials Studio software, which is widely used in materials science and for constructing various 3D molecular models [16]. This software provides greater theoretical support for existing macroscopic experiments and provides a predictive research tool for the implementation of macroscopic experimental protocols, which can reduce the scientific research costs and time costs associated with macroscopic experiments.

Although extensive experimental studies have been conducted to analyze the structure of cellulose, simulation approaches to cellulose studies are relatively rare. This is not surprising because the complexity of cellulose structure makes it difficult to study using traditional molecular simulation methods. A theoretical framework for the study of cellulose polymers was provided by Fukuda et al., who utilized molecular dynamics simulations to examine the diffusion pattern of water molecules in various polymers [17]. In order to model and simulate cellulose, Ftanaka et al. carried out a comparison study of the arrangement of cellulose molecules both with and without moisture [18]. Liao et al. conducted a thorough investigation into the mechanism by which cellulose ages when exposed to moisture. The distribution of hydrogen bonds between molecules was examined under the influence of an increasing temperature and thermal field in a mixed system with a varying moisture distribution. It was found that the diffusion results were unaffected by the distribution of water molecules and that the water molecules would eventually penetrate the amorphous region of cellulose and form hydrogen bonds with other molecules, speeding up the thermal aging of insulating paper [19].

This research examined how varied oxygen concentrations during water vapor heat treatment impacted the mechanical characteristics of wood cellulose. The diffusion coefficient of water molecules, cell volume and density, mean square displacement of cellulose chains, hydrogen bonds, and mechanical parameters were analyzed. Its primary objective is to investigate the changes in macroscopic properties of wood during heat treatment and to give more theoretical support for wood modification.

## 2. Materials and Methods

### 2.1. Modeling

Based on the study of Jiang et al. on the effects of different oxygen concentrations on the properties of heat-treated wood, this paper constructs a microscopic model of the main components of wood using molecular dynamics simulation to explain these changes from a microscopic perspective. And according to the settings of oxygen concentration in the macroscopic experiments by Jiang et al., the number of oxygen and water molecules in six samples were set to 0 and 50, 1 and 49, 2 and 48, 3 and 47, 4 and 46, and 5 and 45, corresponding to oxygen concentrations of 0%, 2%, 4%, 6%, 8%, and 10% in this study [12]. The amorphous polymer building process was used to create the amorhous region of cellulose [20]. In actual applications, the chain length of cellulose is up to several thousand, and the degree of polymerization of several thousand is typically not used due to computer performance and calculation speed. The results of the available studies show that the properties of the material are little affected by the degree of cellulose polymerization [21–23]. Therefore, in order to reduce the simulation time and calculation workload, the cellulose chain with a polymerization degree of 20 was chosen for this study [24]. The model was designed using the software's amorphous cell tool, in which the density was established at 1.5 g/cm$^3$ [25]. Each model contained a wood cellulose molecular chain and a certain number of water and oxygen molecules. The models are shown in Figure 1.

### 2.2. Dynamic Simulation

In the simulation process, the choice of the ensemble and force field is particularly important. The ensemble is a collection of a large number of independent systems with identical properties and structure, in various states of motion, under certain constraints. The cohesion is a fundamental concept introduced by the statistical method to describe the statistical regularity of a thermodynamic system, which is not the actual object but the system that constitutes the cohesion. The constraints are represented by a set of applied macroscopic parameters. In the application of classical molecular dynamics simulation methods, two states of the system—equilibrium and nonequilibrium—can be simulated. In the equilibrium system state, molecular dynamics simulations can be divided into micro-regular system molecular dynamics (NVE) simulations, regular system molecular dynamics (NVT) simulations, isothermal isobaric system molecular dynamics (NPT) simulations, and isoenthalpic isobaric system molecular dynamics (NPH) simulations [26].

The molecular force field is the key core of molecular dynamics simulation, and the appropriateness of the force field selection directly determines the accuracy of the simulation results [23]. The choice of the force field for different atoms is determined by the type of atom, which is one of the bases of simulation. Many force fields, such as PCFF, UFF, Amber, COMPASS, etc., have been developed according to the nature of different atomic structures, transforming the theoretical study of single atoms at the physical level into a deeper chemical study of atoms of many elements of the periodic table. They can calculate various properties of molecules: bond lengths, bond angle torsions, molecular chain motions, mechanical parameters, thermodynamic properties, etc.

To create the system with the least amount of energy, a preliminary geometry optimization based on a 5000-step smart algorithm was first performed when modeling was complete. The PCFF force field was used throughout the study [27]. After the system's energy was minimized and stabilized, the model was simulated at a temperature of 453.15 K under the NPT ensemble with a random beginning velocity and a total simulation length

of 1 ns, which was collected every 5000 steps for analysis. The selection of the simulation step length determines the total simulation time and the accuracy of the results. Too long a step length may cause intense collisions between molecules and lead to data overflow in the system; too short a step length may reduce the ability of the system to search the phase space during the simulation. The temperature control was performed by Nose [28]. The pressure control was carried out by Berendsen [29]. Atom-based analyses were used to determine the Van der Waals force [30]. The Ewald method was employed to calculate the electronic effect [31].

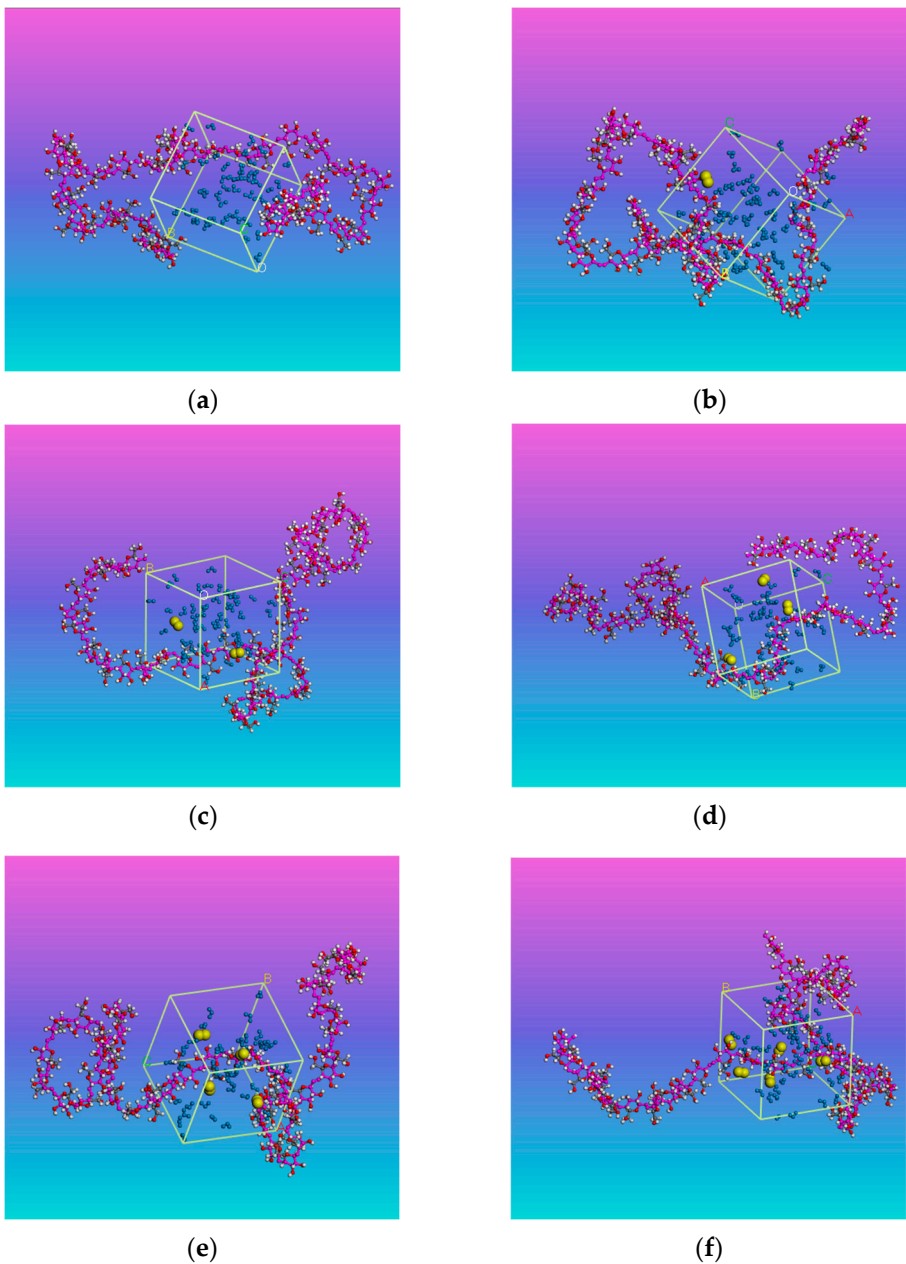

**Figure 1.** Cellulose-water-oxygen model: (**a**) Without oxygen; (**b**) With 2% oxygen; (**c**) With 4% oxygen; (**d**) With 6% oxygen; (**e**) With 8% oxygen; (**f**) With 10% oxygen. The long chain is cellulose chain, the small blue molecules in the cubic cell are water, and the small yellow molecules are oxygen.

## 3. Results and Discussion

### 3.1. Balance of the System

The conditions for determining equilibrium in molecular dynamics simulations are mainly temperature and energy [32,33]. The system is well balanced within a fluctuation interval of 5% to 10%. After simulating the mixed system for 1 ns, a plot of the energy and temperature versus time was obtained for a temperature of 453.15 K, as shown in Figure 2.

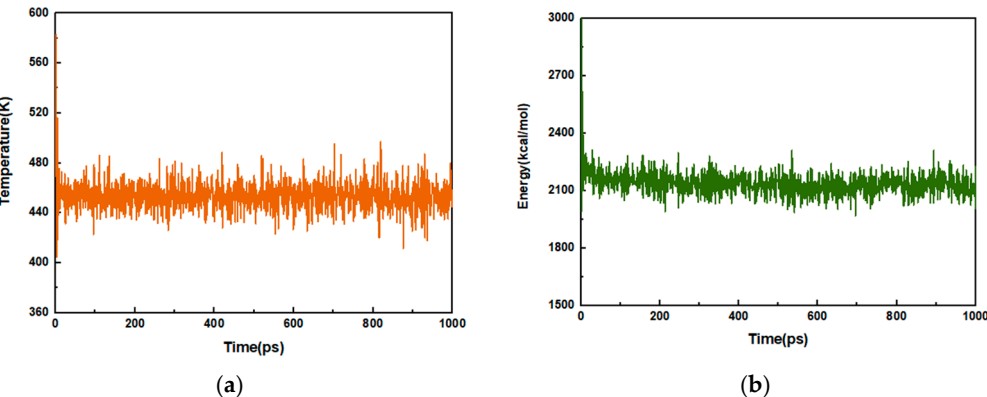

(**a**)           (**b**)

**Figure 2.** Temperature and energy fluctuations of the system: (**a**) Temperature–Time; (**b**) Energy–Time.

Figure 2a shows the temperature fluctuations of the model, whose values are controlled within ±25 K, which indicates that the system is in equilibrium after energy relaxation. Figure 2b shows the energy fluctuations of the model, which range between 2.22% and 3.10% according to the data in Table 1. In summary, the system maintains a good steady state, which proves the reliability of this study.

**Table 1.** Energy Fluctuation Values of the Model at Different Oxygen Concentrations.

| Oxygen Concentration | 0% | 2% | 4% | 6% | 8% | 10% |
|---|---|---|---|---|---|---|
| Fluctuation Value | 2.22% | 2.71% | 3.10% | 2.92% | 2.37% | 2.79% |

### 3.2. Diffusion Coefficient of Water Molecules

The mean square displacement (MSD) is used to observe the positional deviations of the molecules after kinetic simulations, which can represent the motion behavior and migration paths for small molecules in the model. It refers to the number of masses or moles of a substance diffused vertically through a unit area per unit concentration gradient along the direction of diffusion in unit time. The diffusion coefficient is an important part of the study of mass transfer mechanisms. The magnitude of the diffusion coefficient mainly depends on the temperature, the diffusion medium, and the type of diffusing substance, and the basic data are mainly obtained through experiments at present, but the diffusion coefficients under high temperature, high pressure, and adventitious conditions are difficult to obtain through experiments, and the intuitive process of diffusive motion of microscopic particles in the system cannot be observed experimentally. In molecular dynamics simulation (MD), the simulation calculation of diffusion coefficients not only provides microscopic information on the diffusion coefficients of substances and provides some theoretical support for mechanism analysis but also reduces the experimental cost. The MD simulation not only can observe the visual process of particle diffusion motion but also can analyze the trajectory of the particle and obtain the mean squared displacement (MSD) of the particle as a function of time, as expressed by Equation (1),

$$MSD = \sum_{i=1}^{n} \left| \vec{r_i}(t) - \vec{r_i}(0) \right|^2 \tag{1}$$

In the above equation, $\vec{r_i}(t)$ and $\vec{r_i}(0)$ denote the coordinates of the molecule at time t and the coordinates of the molecule at the initial time, respectively. <·> means averaging over all atoms and initial configurations (or time series synthesis). The n represents the number of diffusing particles [13]. In order to make a quantitative comparison of the molecular diffusive motility, the diffusion coefficients (D) were further calculated, which are expressed by Equation (2) [34,35].

$$D = \frac{1}{6n}\lim_{t\to\infty}\frac{d}{dt}\sum_{i=1}^{n}\left|\vec{r_i}(t) - \vec{r_i}(0)\right|^2 \tag{2}$$

when t is large enough, D can be naturally obtained through the slope of the MSD [36], which is expressed by Equation (3),

$$D = m/6 \tag{3}$$

The m in the equation is the MSD curve slope against time obtained after fitting. After analysis, the data for each model are presented in Table 2.

**Table 2.** Diffusion Coefficients of Water Molecules.

| Oxygen Concentration | m | D | R-Square |
|---|---|---|---|
| 0% | 0.4314 | 0.0719 | 0.9980 |
| 2% | 0.5539 | 0.0923 | 0.9987 |
| 4% | 0.5755 | 0.0959 | 0.9989 |
| 6% | 0.6484 | 0.1081 | 0.9978 |
| 8% | 0.6772 | 0.1129 | 0.9995 |
| 10% | 0.7637 | 0.1273 | 0.9993 |

Table 2 demonstrates that the goodness of fit of the water molecule MSD curves are all greater than 0.9, indicating that the results are robust. Figure 3 shows that the D of water molecules gradually rose from 0.0718 to 0.1273, which means that the water molecules in the model possess more flexibility and their movement in the cellulose chain is enhanced. This may be due to the fact that the water molecules possess more kinetic energy as the oxygen concentration increases during the heat treatment.

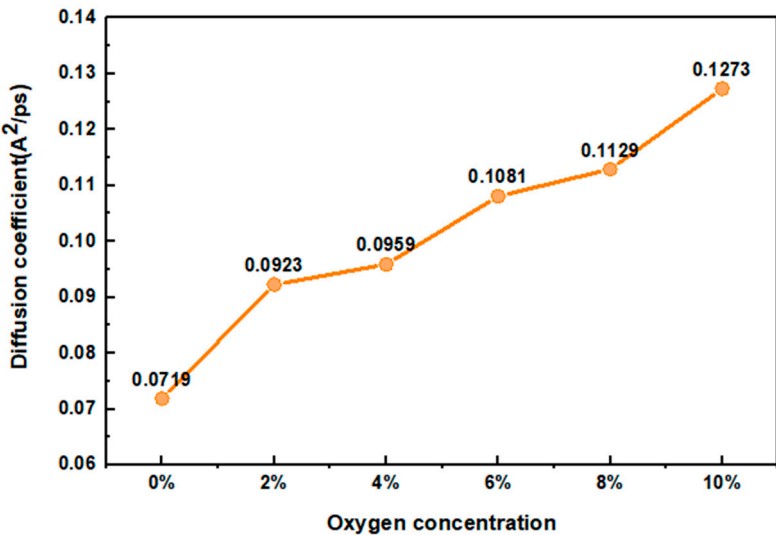

**Figure 3.** Diffusion coefficient of water molecules.

### 3.3. MSD of Wood Cellulose Chains

In the process of studying the effect of hydrogen sulfide on the performance of transformer cellulose insulating paper, Du et al. used the MSD of cellulose chains to represent its movement [37]. The main components of wood and transformer insulating paper are cellulose, so the MSD of wood cellulose can be used to represent the movement of cellulose chains. The degree of violent motion of the cellulose chains determines their stability; the more violent their motion, the less stable they are, thus reducing the overall structural stability of the wood.

Figure 4 reveals that the MSD of the cellulose chains decreased and then rose as the oxygen concentration increased. Compared to the non-oxygen model, the MSD of the cellulose molecular chains was smaller at low oxygen concentrations, indicating that their movement was less displaced and more thermally stable at this time. When the oxygen concentration was greater than 4%, the MSD of wood cellulose chains increased as the oxygen concentration increased. At this time, the motion of cellulose chains was more violent, and the stability and resistance to deformation decreased [38]. This is because the increase in the diffusion coefficient of water molecules accelerates the thermal movement of molecules, making it easier for water molecules to affect the internal structure of cellulose molecular chains, which in turn leads to deformation and cracking of wood [39].

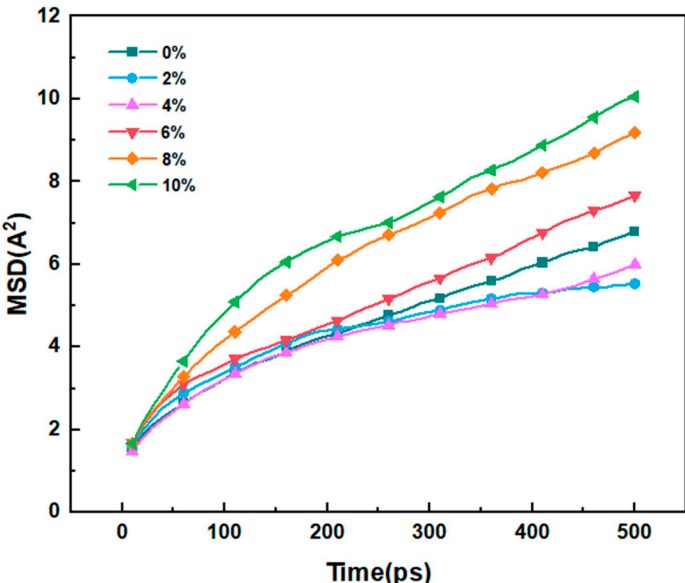

**Figure 4.** Mean square displacement curves of cellulose chains under different oxygen concentrations.

### 3.4. Lattice Parameters and Density

The cell formed by this system is a cube. Its size can be expressed by the lattice parameters. After molecular dynamics simulation, the cell parameters and volume variation of the model in different environments are illustrated in Table 3.

The cell size and density are closely related, and as the cell volume increases, its density decreases accordingly [39]. The density variation of the cellulose-water-oxygen system after water vapor heat treatment at different oxygen concentrations is shown in Table 4.

Where Final and Average represent the final and average values of the model after dynamic simulation, respectively. The final value represents the behavior situation at the last moment, and the average value shows the behavior variation situation throughout the process. The standard deviation (Std. Dev.) is a measure of the dispersion of the data distribution. If the standard deviation is lower, these values deviate less from the mean and are therefore more credible.

**Table 3.** Cell parameter and volume variation of the cellulose-water-oxygen system.

| Oxygen Concentration | Cell Parameters | | | Volume ($A^3$) | | |
|:---:|:---:|:---:|:---:|:---:|:---:|:---:|
| | the Length | the Width | the Height | Final | Average | Std.Dev. |
| 0% | 21.11 | 21.11 | 21.11 | 9404.397 | 9483.821 | 145.470 |
| 2% | 21.16 | 21.16 | 21.16 | 9475.331 | 9535.602 | 121.343 |
| 4% | 21.19 | 21.19 | 21.19 | 9520.700 | 9597.114 | 109.067 |
| 6% | 21.24 | 21.24 | 21.24 | 9577.419 | 9633.354 | 125.889 |
| 8% | 21.31 | 21.31 | 21.31 | 9671.309 | 9655.824 | 115.276 |
| 10% | 21.37 | 21.37 | 21.37 | 9776.236 | 9756.931 | 162.700 |

**Table 4.** Density of the cellulose-water-oxygen system.

| Oxygen Concentration | Density ($g/cm^3$) | | |
|:---:|:---:|:---:|:---:|
| | Final | Average | Std.Dev. |
| 0% | 1.361 | 1.350 | 0.020 |
| 2% | 1.353 | 1.345 | 0.017 |
| 4% | 1.349 | 1.339 | 0.015 |
| 6% | 1.344 | 1.336 | 0.017 |
| 8% | 1.333 | 1.335 | 0.016 |
| 10% | 1.321 | 1.324 | 0.021 |

From the values in Tables 3 and 4, it can be seen that the cell size gradually increased from 21.11 to 21.37 with increasing oxygen concentration, the average volume increased from 9483.821 to 9756.931, and correspondingly, its average density gradually decreased from 1.350 to 1.324. This may be the result of an increasing diffusion coefficient of water molecules. As the flexibility of the water molecules increases, the kinetic energy of the system increases, which in turn leads to an increase in the size of the model and a decrease in density.

*3.5. Hydrogen Bonding*

Hydrogen bonding refers to the powerful non-bonding interaction between a hydrogen atom bonded to an electronegative atom and a neighboring electronegative atom with a lone pair of electrons, which is essentially an electrostatic attraction between a hydrogen nucleus on a strong polar bond and an electronegative atom with a lone pair of electrons [40,41]. A hydrogen bond can be expressed as "X-H...". X is called the donor, Y is called the acceptor, and X-H is called the proton donor. Hydrogen bonds are saturated and directional, and generally, one H-atom may form one or two hydrogen bonds. The mechanical properties of cellulose are greatly influenced by the hydrogen bonding situation. There are numerous hydrogen bonds within and between cellulose molecules [42]. The change of various types of hydrogen bonds at different oxygen concentrations is shown in Table 5.

Table 5 shows that the hydrogen bonding number among cellulose chains first increases and then decreases, indicating that the low oxygen environment enhances the interactions between cellulose chains, which in turn enhances the structural stability. This may be due to the fact that the energy loss during the heat treatment was reduced under the low oxygen concentration, which helped to improve the processing efficiency of the wood heat treatment, resulting in an increase in the number of hydrogen bonds inside the cellulose chains. However, as the oxygen concentration increased, the cellulose molecular chains become more susceptible to the impact of water molecules, leading to a decrease in the number of hydrogen bonds within them. The structure's stability is not only strength-

ened by the formation of hydrogen bonds inside the cellulose chains, but it also alleviates the adverse effects of water molecules on the mechanical properties of cellulose to a certain extent. This also confirms the trend that the MSD of cellulose chains first decreases and then increases. At the same time, the hydrogen bonds between water and cellulose gradually decreased and the interaction between water molecules and cellulose was weakened [43], leading to a weaker adsorption of water molecules by cellulose, which also confirmed the rising tendency of the diffusion coefficient of water molecules.

**Table 5.** Alterations in the various hydrogen bonds.

| Oxygen Concentration | Number of Hydrogen Bonds | | | |
|---|---|---|---|---|
| | between Cellulose Chains | between Water Molecular | between Water–Cellulose | Total |
| 0% | 73 | 30 | 91 | 194 |
| 2% | 96 | 29 | 90 | 215 |
| 4% | 91 | 25 | 81 | 197 |
| 6% | 87 | 28 | 75 | 190 |
| 8% | 90 | 21 | 70 | 181 |
| 10% | 83 | 22 | 70 | 175 |

### 3.6. Mechanical Properties

Mechanical properties are one of the important properties of the material, directly affecting the processing and production of materials and their safe use. This is an important performance related to wood processing, production, and use. According to the principle of elastic mechanics [44], when the material is subjected to external forces, the most common connection of stress and strain satisfies the generalized Hooke's law. For a completely isotropic body, there are only two independent elastic coefficients, $C_{11}$ and $C_{12}$. The reason is that the correlation of the elastic coefficients increases with the symmetry between the bodies, and their stress-strain behavior can be characterized through two individual constants, such that $C_{12} = \lambda$ and $C_{11} - C_{12} = \mu$. The rigidity matrix can be written as Equation (4),

$$[C_{ij}] = \begin{bmatrix} \lambda+2\mu & \lambda & \lambda & 0 & 0 & 0 \\ \lambda & \lambda+2\mu & \lambda & 0 & 0 & 0 \\ \lambda & \lambda & \lambda+2\mu & 0 & 0 & 0 \\ 0 & 0 & 0 & \mu & 0 & 0 \\ 0 & 0 & 0 & 0 & \mu & 0 \\ 0 & 0 & 0 & 0 & 0 & \mu \end{bmatrix} \tag{4}$$

where $\lambda$ and $\mu$ are known as the Lamé coefficients, which serve to derive physical quantities, including Young's modulus (E), bulk modulus (K), shear modulus (G), and Poisson's ratio ($\gamma$).

$$E = \frac{\mu(3\lambda+2\mu)}{\mu+\lambda} \tag{5}$$

$$G = \mu \tag{6}$$

$$K = \lambda + \frac{2}{3\mu} \tag{7}$$

$$\gamma = \frac{\lambda}{2(\lambda+\mu)} \tag{8}$$

where E is the ratio of tensile stress to tensile strain in the elastic deformation of the material and is used to measure the stiffness of the material; a larger value means greater resistance to deformation. Where $\gamma$ is the proportion of lateral to lengthwise deformation during material stretching, which can reflect the plasticity. K/G denotes the proportion of the

bulk modulus to the shear modulus that measures a material's resilience. The higher its value, the greater the material's resiliency [45]. Through MD calculations, the mechanical properties of each hybrid model at different oxygen concentrations were obtained, as presented in Table 6.

**Table 6.** Analysis of Mechanical Parameters.

| Oxygen Concentration | λ | μ | G (GPa) | E (GPa) | γ (GPa) | K/G |
|---|---|---|---|---|---|---|
| 0% | 7.15 | 5.66 | 5.66 | 14.48 | 0.28 | 1.28 |
| 2% | 8.24 | 8.28 | 8.28 | 20.70 | 0.25 | 1.00 |
| 4% | 10.13 | 6.52 | 6.52 | 17.01 | 0.30 | 1.5 |
| 6% | 5.41 | 6.52 | 6.52 | 15.99 | 0.23 | 0.85 |
| 8% | 4.47 | 4.95 | 4.95 | 12.26 | 0.24 | 0.93 |
| 10% | 3.70 | 4.03 | 4.03 | 9.99 | 0.24 | 0.96 |

Table 6 shows that the values of γ and K/G of cellulose chains did not change significantly as oxygen concentration increased. However, it can be seen from Figure 5 that E and G showed an obvious trend of first increasing and then decreasing, indicating that the deformation resistance and rigidity of cellulose chains first strengthened and then weakened. At the oxygen concentration of 2%, the shear modulus of cellulose chains was 8.28 and Young's modulus was 20.70, both of which reached maximum values. These results are consistent with the conclusion that water vapor heat treatment under the appropriate oxygen concentration can improve the flexural elastic modulus and stiffness of wood [12]. In addition, the trends of E and G are in good agreement with the trends of MSD and hydrogen bonding of cellulose chains, indicating that the internal structure of cellulose chains is relatively stable at low oxygen concentrations, but that its internal structure is damaged and becomes increasingly unstable at high oxygen concentrations.

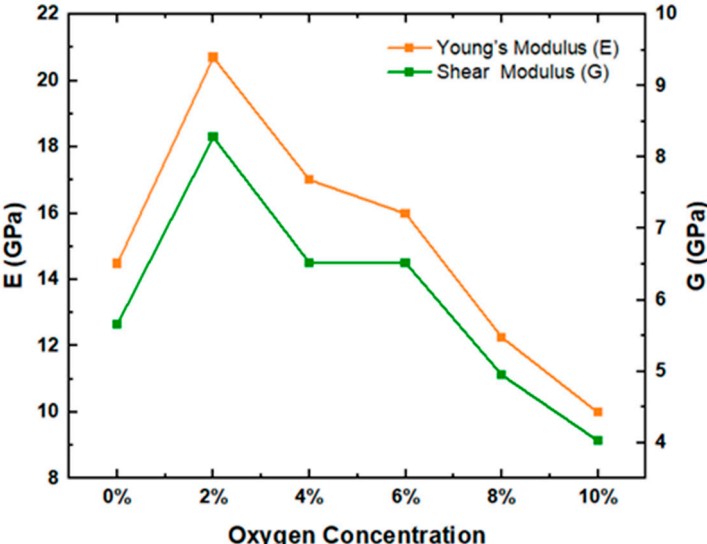

**Figure 5.** Young's modulus and shear modulus in the model.

There are many methods of modification of wood heat treatment, and previous studies have analyzed the effects of medium, temperature, and pressure on the mechanical properties of wood from macroscopic and microscopic perspectives. In this paper, the effect of oxygen concentration on the mechanical properties of wood during water vapor heat treatment was studied for the first time from the microscopic perspective of molecular

dynamics simulation. The results demonstrate that an appropriate increase in oxygen concentration can significantly enhance the rigidity and deformation resistance of wood. This provides more theoretical support for wood modification at the microscopic level, which can better meet the requirements of wood rigidity in various industries.

## 4. Conclusions

This research examined how varied oxygen concentrations during water vapor heat treatment impacted the mechanical characteristics of wood cellulose. The diffusion coefficient of water molecules, cell volume and density, mean square displacement of cellulose chains, hydrogen bonds, and mechanical parameters were analyzed. The following conclusions are obtained:

1. Water molecules become more flexible in the presence of oxygen, which causes the diffusion coefficient of water molecules to gradually rise with oxygen concentration. This aids in minimizing energy loss during heat treatment, thus improving the processing efficiency. At the same time, the increase in water molecule diffusion coefficient causes a simultaneous rise in cell volume and a corresponding fall in density.
2. The MSD of cellulose chains decreases and then increases with the increase in oxygen concentration, which indicates that the thermal stability of cellulose chains is better at low oxygen concentrations, and oxygen concentrations that are too high will lead to the destruction of the internal structure of cellulose chains and greatly reduce the stability. This is related to the number of hydrogen bonds within the cellulose chain. The formation of intermolecular hydrogen bonds increases the molecular interactions within the cellulose chains and enhances the stability of the structure.
3. Young's modulus and shear modulus of cellulose chains first rise and then decline with oxygen concentration, indicating that the rigidity and distortion resistance of cellulose chains improve and then fall, peaking at 2% oxygen concentration. This change trend is compatible with the cellulose chains' hydrogen bonding and MSD change trends, which further indicate that the internal structure of cellulose chains is more stable at low oxygen concentrations. These results indicate that an appropriate increase in oxygen concentration can help to potentially improve the stiffness and resistance to deformation of wood, and also confirm the significance of this paper's research, which provides additional theoretical support for the development of wood heat treatment processes.

**Author Contributions:** Conceptualization, Y.G.; methodology, Y.G.; software, Y.G.; validation, Y.G., W.W. and X.J.; formal analysis, Y.G.; investigation, X.J.; resources, Y.G.; data curation, Y.G.; writing—original draft preparation, Y.G.; writing—review and editing, Y.G.; visualization, Y.G.; supervision, X.J.; project administration, Y.G.; funding acquisition, W.W. All authors have read and agreed to the published version of the manuscript.

**Funding:** This research was funded by the Fundamental Research Funds for the Central Universities, grant number 2572019BL04, and the Scientific Research Foundation for the Returned Overseas Chinese Scholars of Heilongjiang Province, grant number LC201407.

**Institutional Review Board Statement:** Not applicable.

**Informed Consent Statement:** Not applicable.

**Data Availability Statement:** Data are available upon request from the corresponding author.

**Conflicts of Interest:** The authors declare no conflict of interest.

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
