# Peer review of "Molecular Dynamics Study on Mechanical Properties of Cellulose with Water Molecules Diffusion Behavior at Different Oxygen Concentrations"

_forests, doi:10.3390/f14020371_

Round 1
Reviewer 1 Report
The article is interesting, however the discussion of the results is almost non-existent

Reviewer 2 Report
The topic of the research work and manuscript is really interesting and provides new information. However there are several issues to be addressed towards its quality improvement. Please check my comments to the authors and I remain at your disposal for any clarification.
In line 27, the phrase "convenient transportation" needs improvement. In line 30, please provide the relevant reference entitled "Effect of thermal treatment on colour and hygroscopic properties of poplar wood" in the end of sentence. In lines 42-43, please provide more information about the referred "appropriate range". In line 46, explain the "under low oxygen concentration", as well as the "high oxygen concentrations" in line 47. In lines 50 and 57, provide some references (the total number of references 28 could be considered rather low for such a topic with rich literature). The term "celluloses" should rather be used in single form. In line 65, are these research processes so expensive or rather time-consuming to be highlighted. The last sentence of introduction should be rephrased, it is not clear, as well as is it a question or suggestion to the reader? In the introduction, the rest of thermal treatment induced changes in chemical constituents of wood are not at all referred, not even the changes in cell structure (since the anatomical characteristics of thermal wood are closer to the topic of this article/research work), not even respective modelling approaches published on this topic so far. Generally in the text, you do not make a separation between the changes of hardwoods and softwoods (are there any significant differences?this should probably be referred). In figure 1 caption, the colours should be as well explained. Is there any specific wood species being examined in the simulation process? Please explain in detail in the materials-methods chapter how did you validate the modelling application findings in this work, since it is not very clear in the text. In line 252, do you refer during the treatment? In 254 line, the "the more the number" should be totally improved. In 262, again, do you mean during the treatment? What is the practical meaning and significance of this work's findings and the future utilization potential of them should be further analyzed in the discussion chapter and just highlighted in the conclusions chapter.
Round 2
Reviewer 1 Report
The work improved after the authors' review and can be published
Author Response
Thank you again for your hard work and your suggestions are very helpful to us. I hope your work goes well and you live a happy life.
Reviewer 2 Report
Unfortunately, the authors have implemented only few of the proposed changes, corrections and additions, and they have not answered adequately in their response to the points of the review. There is not any convincing answer about the validation process, the statistical analysis. The recommended reference has not been referred, and the discussion chapter continues to be quite poor/not properly discussed. They have to check the review points one by one and make the necessary corrections and additions. Furthermore, in the added parts of text grammatical and syntactical errors have been detected.
